# Modulation of Type 5 Metabotropic Glutamate Receptor-Mediated Intracellular Calcium Mobilization by Regulator of G Protein Signaling 4 (RGS4) in Cultured Astrocytes

**DOI:** 10.3390/cells13040291

**Published:** 2024-02-06

**Authors:** Pauline Beckers, Pierre J. Doyen, Emmanuel Hermans

**Affiliations:** Institute of Neuroscience, Université Catholique de Louvain, 1200 Brussels, Belgium; pauline.beckers@uclouvain.be (P.B.); pierre.doyen@unamur.be (P.J.D.)

**Keywords:** regulators of G protein signaling, metabotropic glutamate receptor, calcium oscillations, astrocytes, glial cells

## Abstract

Acting as GTPase activating proteins promoting the silencing of activated G-proteins, regulators of G protein signaling (RGSs) are generally considered negative modulators of cell signaling. In the CNS, the expression of RGS4 is altered in diverse pathologies and its upregulation was reported in astrocytes exposed to an inflammatory environment. In a model of cultured cortical astrocytes, we herein investigate the influence of RGS4 on intracellular calcium signaling mediated by type 5 metabotropic glutamate receptor (mGluR5), which is known to support the bidirectional communication between neurons and glial cells. RGS4 activity was manipulated by exposure to the inhibitor CCG 63802 or by infecting the cells with lentiviruses designed to achieve the silencing or overexpression of RGS4. The pharmacological inhibition or silencing of RGS4 resulted in a decrease in the percentage of cells responding to the mGluR5 agonist DHPG and in the proportion of cells showing typical calcium oscillations. Conversely, RGS4-lentivirus infection increased the percentage of cells showing calcium oscillations. While the physiological implication of cytosolic calcium oscillations in astrocytes is still under investigation, the fine-tuning of calcium signaling likely determines the coding of diverse biological events. Indirect signaling modulators such as RGS4 inhibitors, used in combination with receptor ligands, could pave the way for new therapeutic approaches for diverse neurological disorders with improved efficacy and selectivity.

## 1. Introduction

Astrocytes play a crucial role as pivotal contributors to the regulation of neuronal glutamate transmission. Best documented is their capacity to ensure the rapid clearance of neuronally released glutamate from the synaptic cleft through efficient uptake and metabolism, which is indispensable to control excitatory transmission and prevent excitotoxicity [1,2]. However, the influence of astrocytes extends beyond the conventional role of high-affinity glutamate transporters as they also release glutamate through the cystine–glutamate exchanger (the system x_c_^−^ senses extracellular glutamate through specific metabotropic glutamate receptors (mGluRs), a G-protein-coupled receptor family (GPCR). Subtypes 3 and 5 mGluR are expressed in astrocytes with a prominent role for the latter in bidirectional communication between neurons and glial cells, which appears essential for the regulation of synaptic function. In response to changes in the local excitatory tone, mGluR5 modulates the astrocytic release of gliotransmitters, such as ATP, neurotrophic factors, d-serine, glutamate and GABA, which in turn influence neuronal responses [3,4,5,6,7]. Hence, documented alterations in astroglial mGluR5 signaling or expression have been observed in models of neurological disorders such as amyotrophic lateral sclerosis (ALS), chronic neuropathic pain and Alzheimer’s disease [8].

mGluR5 preferentially couples to G_q/11_-type G-proteins, thereby triggering intracellular calcium mobilization in response to synaptic glutamatergic activity [9,10]. Depending on the cell density, receptor expression levels or functional interactions with signaling partners, single-peak responses, peak–plateau responses or Ca^2+^ oscillations have been reported both in astrocytes and in transfected cells expressing recombinant mGluR5 [11,12,13,14]. It has been suggested that calcium signal complexity supports differential regulation of down-stream effectors [15,16], and the nature of the calcium signal triggered by glutamate or agonists of mGluR5 in astrocytes could contribute to the fine-tuning of the synaptic activity.

Amongst the different partners that influence GPCR signaling, the so-called regulators of G protein signaling (RGSs) have received increasing attention, particularly since the discovery of several small, specific inhibitors holding great promise for future developments in pharmacology [17,18]. RGSs comprise a large family of small intracellular proteins initially identified as GTPase accelerating protein (GAP) that ensure rapid silencing of activated G-proteins. As such, RGSs are generally considered negative modulators of cell signals, but these proteins have also been assigned other functions through interactions with other signaling partners or through their scaffold properties [19,20,21]. The expression level of RGSs is thought to determine their impact on GPCR signaling and the accumulating data demonstrate that a variety of mechanisms regulate RGSs in physiological and pathological contexts [22].

Amongst the large diversity of RGSs identified so far, RGS2, RGS4, RGS7, RGS9 and RGS14 appear enriched in nervous tissues with demonstrated roles in modulating synaptic signaling and plasticity [23]. Hence, RGS4 is documented as one of the predominant members in the central nervous system [24,25] and has received much attention, with repeated demonstrations of its dysregulation in several neurological or psychiatric disorders such as schizophrenia, Alzheimer disease, drug addiction, brain tumors and neuropathic pain [26,27,28,29,30]. Hence, targeting RGS4 has been proposed as a promising therapeutic approach in some of these diseases [31,32,33,34].

In relation to mGluR5, early studies conducted by Saugstad et al. in 1998 demonstrated that RGS4 plays a role in attenuating the response to the selective agonist dihydrophenylglycine (DHPG) in hippocampal neurons [35]. Subsequent research by Schwendt et al. in 2012 confirmed a functional interaction between RGS4 and mGluR5 in neurons [36]. Their study revealed that RGS4 overexpression in the dorsal striatum suppressed both behavioral activity and ERK phosphorylation induced by mGluR5 inactivation [36]. Apart from studies on RGS10 in microglia and more recent evidence for RGS5 in astrocytes, the expression, regulation and function of RGSs in glial cells remain, so far, poorly investigated [37,38]. RGS4 expression is documented in both neurons and glial cells, and we have shown that mimicking neuroinflammation using soluble inflammatory mediators could alter RGS4 expression in cultured astrocytes [39]. Nevertheless, the functional consequence of an interaction between RGS4 and mGluR5 in astrocytes has never been examined. Considering the importance of calcium signaling in the regulation of numerous physiological and pathological processes by astrocytes, we herein analyze the regulation of mGluR5-mediated intracellular Ca^2+^ mobilization after manipulating RGS4 expression or activity. Modifications in the mGluR5 signaling profile, influenced by molecular partners such as RGSs, could potentially contribute to alterations in the crosstalk between astrocytes and neurons during excitatory transmission in the tripartite synapse.

## 2. Materials and Methods

### 2.1. Astrocyte Cultures

On postnatal day 2, rat pups were sacrificed and the cortex was isolated by dissection. The hippocampus and meninges were removed and the cortical gray matter was dissociated in PBS-glucose 0.2%. Astrocytes were then separated from other cells using a Percoll 30% gradient (GE Healthcare, Chicago, IL, USA). Cells were finally washed in PBS-glucose and seeded in gelatin-coated 175 cm^2^ flasks. Astrocytes were left to proliferate at 37 °C in a humidified atmosphere containing 5% CO_2_ in Dulbecco’s modified Eagle’s medium (glutaMAX, Thermo Fisher Scientific, Waltham, MA, USA) supplemented with 10% fetal bovine serum (FBS) (Thermo Fisher Scientific, Waltham, MA, USA), 50 mg/mL penicillin-streptomycin (Thermo Fisher Scientific, Waltham, MA, USA) and 50 mg/mL fungizone (Thermo Fisher Scientific, Waltham, MA, USA) for two weeks. The medium was renewed after one week. On day 15, cells were trypsinized and seeded into multi-well plates for 2 days in medium supplemented with 10% of FBS. On day 17, the serum concentration was decreased to 3% and, when indicated, the medium was supplemented with the G5 supplement, a growth factor cocktail (Thermo Fisher Scientific, Waltham, MA, USA).

### 2.2. Immunocytochemistry

Astrocytes were seeded in 24-well plates. After 7 days of maturation in 3% FBS-medium supplemented or not with G5 supplement, the cells were washed three times in phosphate-buffered saline (PBS). Then, they were fixed with paraformaldehyde 4% in PBS for 30 min on ice. After further washing, the cells were permeabilized with Triton X-100 1% (Pharmacia, Uppsala, Sweden) in PBS. Blocking was performed with bovine serum albumin 1% in PBS and the cells were incubated overnight at 4 °C with the polyclonal rabbit anti-mGluR5 primary antibody (1/1000, AB5675, Merck, Rahway, NJ, USA) or the monoclonal mouse Cy3-coupled anti-glial fibrillary acidic protein (GFAP) antibody (1/500, C9205, Merck). The next day, the astrocytes were rinsed 3 times with PBS and were incubated for 1 h with the secondary antibody goat anti-rabbit Alexa fluor 488 (1/500, Thermo Fisher Scientific). Finally, the cell nuclei were stained to 4′,6-diamidino-2-phenylindole (DAPI) at 0.2 µg/mL (Merck) in PBS. Fluoprep (bioMérieux SA, Marcy-l’Étoile, France) was used as a mounting medium and the cells were analyzed using an Evos FL Digital Inverted Microscope (Westburg, Pune, Maharashtra, India).

### 2.3. Monitoring of Intracellular Ca^2+^ Mobilization

Changes in cytosolic Ca^2+^ concentration were examined in individual astrocytes using calcium-sensitive fluorescent dye Fura2. Astrocytes were grown on poly-L-lysine-coated (Merck, Darmstadt, Germany) 15 mm diameter glass coverslips (Marienfeld-Superior, Lauda-Königshofen, Germany) and incubated with 5 µM Fura2-acetoxymethyl ester (Thermo Fisher Scientific, Waltham, MA, USA) for 40 min in a Krebs buffer (118 mM NaCl, 4.7 mM KCl, 4 mM NaHCO_3_, 1.2 mM MgSO_4_, 1.2 mM KH_2_PO_4_, 8.5 mM Hepes, 1.3 mM CaCl_2_, 11.7 mM glucose, pH 7.4). Coverslips were rinsed and mounted on a heated (37 °C) and perfused microscope chamber (Warner Instrument Corporation, Holliston, MA, USA). While being continuously perfused with heated Krebs buffer, Fura2-loaded astrocytes were excited successively at 340 and 380 nm (excitation light was obtained from a xenon lamp coupled to a monochromator) for 2 × 100 ms and the emitted fluorescence was monitored at 510 nm using a charged device sensor camera coupled to an inverted Olympus IX70 microscope (TILL photonics). Fluorescence intensities from every single astrocyte in the defined field (between 7 and 25 cells for each experiment) were recorded separately at a frequency of 1 Hz, corrected for background and combined (fluorescence ratio F340/F380) using the software TILLvisION version 3.3. Cytosolic Ca^2+^ concentration changes expressed as a fluorescence ratio F340/F380 were monitored upon application of (S)-3,5-dihydroxyphenylglycine ((S)-3,5-DHPG, mGluR agonist, Tocris, Bristol, UK) for 120 s and 3-((2-methyl-4-thiazolyl)ethynyl)pyridine (MTEP, mGluR5 antagonist, Hello Bio, Princeton, NJ, USA) for 60 s. In some experiments, the decay in the Fura2 signal ratio was monitored using a linear regression analysis between 30 and 90 s after the application of DHPG.

### 2.4. Western Blot

Astrocytes seeded in 6-well plates were rinsed with PBS and scraped in ice-cold Laemmli sample buffer (2×) (Tris 125 mM pH6.8, glycerol 20%, SDS 4%) in the presence of cOmplete™ protease inhibitor cocktail (Merck, Darmstadt, Germany) and sonicated. Samples were centrifuged at 1000× *g* for 3 min to remove insoluble material. Protein concentration was determined using the DC Protein Assay Reagents Package and samples were diluted in a loading buffer (TrisHCl 125 mM pH 6.8, glycerol 20%, sodium dodecyl sulfate (SDS) 4% and bromophenol blue 0.01%) and boiled for 5 min. Samples were electrophoresed through 10% SDS-PAGE and blotted to a nitrocellulose membrane using an Owl™ HEP-1 semidry electroblotting system (Thermo Fisher Scientific, Waltham, MA, USA) in a transfer buffer (Tris 48 mM, glycine 39 mM, SDS 0.037%, methanol 20%, pH 8.3). Immunoprobing was carried out by incubating membranes overnight at 4 °C with rabbit anti-RGS4 primary antibodies (15129S, 1/800, Cell Signaling Technology, Danvers, MA, USA), mouse anti-mycTag (46-0603, 1/1000, Thermo Fisher, Waltham, MA, USA), mouse anti-β-Tubulin (556321, 1/5000, BD Biosciences, Dubai, United Arab Emirates) and rabbit anti-GAPDH antibodies (G9545, 1/30000, Merck, Darmstadt, Germany). Membranes were then incubated for 1 h at room temperature with peroxidase-conjugated anti-rabbit IgG (1/3000) or goat anti-mouse IgG (1/3000) secondary antibodies. Immunoreactive proteins were detected with an enhanced chemiluminescence reagent followed by autoradiography.

### 2.5. MTT Assay

After two weeks of proliferation, the astrocytes were plated in 96-well plates (5000 cells/well) for one week and treated with 1 to 100 µM of CCG 63802 (RGS4 inhibitor, Tocris, Bristol, United Kingdoms) for 40 min. At this stage, some cells were grown in a fresh culture medium for 3 additional days in a humidified atmosphere with 5% CO_2_ at 37 °C. Then, cells were washed in Hank’s balanced salt solution (HBSS) containing Ca^2+^/Mg^2+^ (Lonza, Bâle, Switzerland, 10-527F) supplemented with 10 mM of Hepes (Gibco, New York, NY, USA, 15630) and incubated for 3 h with a saturated MTT (Calbiochem, San Diego, CA, USA, 475989) solution in the washing buffer in a humidified atmosphere with 5% CO_2_ at 37 °C. A H_2_O_2_ concentration of 30% (*w*/*w*) for 10 min was used as a positive control. The formazan deposit was solubilized in 100 µL 2-propanol/HCl 0.04 N. Absorbance at 570 nm was measured with a SpectraMax i3 plate imager (Molecular Devices, San Jose, CA, USA).

### 2.6. Construction of Lentiviral Vectors

RGS4 overexpression: A COOH-terminal myc-6xHis-tagged RGS4 cDNA sequence inserted previously [32] in a pcDNA5FRT/TO vector (Thermo Fisher, Waltham, MA, USA) was amplified with two specific primers, with one of them harboring a supplementary XmaI restriction site. The sequence was then inserted in a pcDNA™3.1/V5-His TOPO™ plasmid (Invitrogen, Waltham, MA, USA) and further cloned in the bicistronic pTM945 vector coding for an mCherry fluorescent protein [40] using B*am*HI and X*ma*I restriction enzymes. The cloned myc-6xHis-RGS4 sequence was validated by double-strand sequencing. The original pTM945 vector without any insertion was used as a negative control. Lentiviral particles were generated by the transient transfection of 293T cells grown in 175 cm^2^ flasks using 26.17 µg of lentiviral backbone (further abbreviated pTM945-RGS4), 4.36 µg of pREV, 5.23 µg of pVSVg and 8.72 µg of pMDL and the TransIT^®^-293 transfection reagent (Mirus Bio, Madison, WI, USA). The transfection efficiency was validated by the detection of mCherry-positive cells. After 48 h at 37 °C in a humidified atmosphere with 5% CO_2_, cell supernatants were harvested, centrifuged at 1500 and passed through a 0.8/0.2 µm Acrodisc^®^ Syringe filter (Pall corporation, New York, NY, USA). The virus suspension was concentrated 200 times after centrifugation at 1500× *g* for 45 min at 4 °C in the presence of Lenti-X™ Concentrator (Takara Bio, Kusatsu, Japan).

RGS4 down expression: A short hairpin RNA (shRNA) sequence targeting the rat RGS4 (GAAGTCAAGAAATGGG) transcript was selected using RNA mission collection (Merck). They were cloned in pMK117 [41] using B*sh*TI and E*co*RI. This construct also harbors the gene encoding for the fluorescent protein E2-Crimson, facilitating the detection of infected cells. The same construct containing a scrambled sequence with the same nucleotides was used as a negative control. Lentiviral particles were produced by the transfection of 293T cells with 15 µg of lentiviral vectors, 11.25 µg of pSPAX2 (Addgene, Cambridge, MA, USA) and 4.5 µg of pMD2-VSV-G. Lentiviral particles in the supernatant were collected 48 h post transfection, passed through a 0.45 µm filter and concentrated using Vivaspin columns (cutoff: 1,000,000 Da; Vivaspin, Sartorius, Göttingen, Germany). The produced lentiviral particles were titrated by flow cytometry and by transducing HEK293 cells with dilutions of lentiviruses followed by the quantification of E2-Crimson-positive cells. The inhibition of RGS4 expression in astrocytes was assessed using serial dilutions of the final suspension of lentiviruses and the 4× dilution was selected for calcium signaling experiments.

### 2.7. Statistical Analyses

Data were obtained from at least three independent experiments and were expressed as means with the standard error of the mean (SEM). Graphics and descriptive statistics were elaborated using GraphPad Prism 5.03 (GraphPad Software, Boston, MA, USA). Statistical analyses were conducted using R, version 4.3.1 (16 June 2023). As mentioned in the figure legends, a hierarchical mixed logistic regression (fitted by maximum likelihood) with “treatment” as fixed and experiment repetition as the random effect, along with Wald *z*-tests for fixed effects comparison or one-way ANOVA followed by Student pairwise comparisons, were conducted. In all statistical analyses, a value of *p* < 0.05 was considered significant.

## 3. Results

### 3.1. The Astrocyte Maturation Protocol Influences the Expression and the Function of mGluR5

The expression and the functional properties of mGluR5 in primary cultured astrocytes were examined using maturation conditions consisting of decreasing the serum concentration to 3% and adding or not a standardized combination of growth factors (supplement G5) for one week. While astrocytes maintained in the FBS 3% condition adopted a protoplasmic phenotype, cells exposed to the culture supplement G5 adopted the star shaped morphology of stellate astrocytes (Figure 1A). In astrocytes grown in FBS 3%-medium, immunodetection of the astroglial marker GFAP revealed a signal that appeared to be condensed in the cell body, whereas intense GFAP staining was detected throughout the extended cell processes of G5-matured astrocytes. Consistent with previous studies examining mGluR5 mRNA levels [42], the immunodetection of mGluR5 revealed an intense signal in stellate astrocytes within the cell processes which is not observed in protoplasmic astrocytes. We further analyzed the calcium signaling profile triggered by the activation of mGluR5 upon exposure to the group I selective agonist DHPG. Both protoplasmic and stellated astrocytes showed robust increases in intracellular Ca^2+^ concentration upon application of 50 µM of DHPG. However, after the initial peak, the response profiles appeared to be considerably different. In protoplasmic cells, the initial peak was followed by a sustained and progressive decrease (Figure 1B). This typical peak–plateau-type response was observed in nearly 90% of the cells, while none of the cells showed intracellular Ca^2+^ oscillations (Figure 1D). In contrast, stellate astrocytes predominantly showed repeated Ca^2+^ oscillations in response to DHPG (Figure 1C,D). In both models, the intracellular Ca^2+^ concentration shortly returned to basal levels after the addition of the specific mGluR5 antagonist MTEP (10 µM).

### 3.2. RGS4 Inhibition Modulates mGluR5-Associated Calcium Signaling

The cell signaling associated with astrocytic mGluR5 was analyzed after exposure of G5-matured cultures to the RGS4 specific inhibitor CCG 63802 [43]. The potential toxicity of this inhibitor was first investigated in an MTT assay or by quantifying its influence on cell density (Figure 2A–C). The 40 min duration of exposure to the RGS4 inhibitor, defined in line with the protocol adopted for the calcium mobilization experiments, did not cause any noticeable mitochondrial impairment in astrocytes. However, assuming that the consequence of early toxicity might only be detected at a later time point, the assay was also conducted 3 days after the initial 40 min treatment. In these conditions, a substantial decrease in the mitochondrial activity was only observed with the highest concentration of the RGS inhibitor tested (100 µM), but not with the lowest ones (10 and 30 µM). Notably, this toxicity revealed by a loss in MTT metabolizing activity did not translate into a significant decrease in cell density after 3 days. Subsequent experiments were only performed with cells exposed to 10 and 30 µM of CCG 63802 and the influence of this compound was examined on mGluR5-associated calcium signaling triggered by DHPG in G5-matured astrocytes (Figure 2D–F). Cultures exposed to 10 or 30 µM CCG 63802 showed a reduced percentage of responding cells as well as a decrease in the proportion of cells showing Ca^2+^ oscillations (Figure 2G,H). The proportion of cells showing a peak–plateau-type response was not significantly changed. In all responding cells, the amplitude of the first peak was also negatively affected by the inhibitor (Figure 2I).

### 3.3. RGS4 Overexpression Modulates mGluR5-Associated Calcium Signaling

Considering the low efficiency of transfection in astrocytes using conventional protocols, a lentiviral-based approach was used to increase RGS4 expression in G5-matured astrocytes. We designed lentiviruses coding for the myc-6xHis tagged full-length RGS4 protein as well as for the fluorescent protein mCherry. The same lentiviruses lacking the RGS4 insert were used as a negative control. The evaluation of mCherry-associated fluorescence revealed a high infection efficiency with the detection of more than 90% of positive cells, both for the control and RGS4-encoding viruses, while no fluorescence signal was observed in non-infected cells (Figure 3A). The expression of the recombinant RGS4 in the infected cultures was also confirmed by immunoblotting experiments using an myc-detecting antibody (Figure 3B).

Infecting astrocyte cultures modestly influenced the calcium signaling induced by DHPG. Thus, approximately 45% of the cells from the cultures infected with the control lentivirus (lacking the RGS4 insert) responded to DHPG with an oscillatory calcium profile, similar to the proportion found in non-infected cells (see Figure 2H). The overexpression of recombinant RGS4 with the designed lentiviruses increased the percentage of oscillating cells to nearly 80% and this was accompanied by a decrease in the percentage of cells showing a peak–plateau-type response (Figure 3C–E). When specifically examining the cells responding with a peak–plateau profile, we determined that the slope of the Ca^2+^ concentration decay (plateau-phase) was significantly higher after RGS4 infection, indicating a faster decrease in the cytosolic Ca^2+^ concentration during the exposure to DHPG (Figure 3F–H).

### 3.4. Knockdown of RGS4 Modulates mGluR5-Associated Calcium Signaling in Astrocytes

In complement to the pharmacological approach aimed at decreasing RGS activity, shRNA lentiviruses were used to reduce the expression of RGS4 in G5-matured astrocyte cultures. As for the experiments aiming at overexpressing RGS4, the efficiency of the viral infection was confirmed by examining the fluorescence signal associated with the expression of the E2-Crimson protein driven by the lentivirus constructs. Infection with both the scrambled-shRNA lentivirus and the RGS4-shRNA lentivirus resulted in a substantial long-red fluorescent signal in more than 90% of the astrocytes (Figure 4A). The knockdown of RGS4 was verified by immunoblotting using serial dilutions of the lentivirus suspension, confirming a considerable decrease in the signal corresponding to RGS4 (Figure 4B). Hence, we previously validated the use of this RGS4-shRNA lentivirus in astrocyte cultures [32]. After infection of the cells with the scrambled-shRNA lentivirus, the majority of the cells responded to DHPG with an oscillatory calcium signaling profile, as previously shown with non-infected cells (see Figure 2H). At variance, the reduction in RGS4 expression obtained using the RGS4-shRNA lentivirus resulted in a profound change in the calcium signaling associated with mGluR5 activation as most of the cells adopted a peak–plateau-type response (Figure 4C–E).

## 4. Discussion

Recognized as the major excitatory neurotransmitter, glutamate also appears as a potent neurotoxin in diverse pathological situations. The pharmacological adjustment of glutamate levels in the central nervous system is frequently proposed as a conceivable option in the treatment of neurological disorders. Whereas the activation of postsynaptic ionotropic receptors perpetrates the action potential, the activation of mGluRs triggers slower and more persistent modifications that can modulate neurotransmission and influence synaptic plasticity [44,45]. mGluRs are localized at both pre- and postsynaptic terminals [46] but their presence in glial cells is well documented [9]; they can modulate several activities, including cell proliferation [47], glutamate uptake [42,48] and inflammatory responses [49,50]. mGluR5 is commonly reported as the single group I mGluRs expressed in glial cells, in particular on astrocytes [48,49], where it regulates the glial activities in responses to excitatory synaptic activity [9]. Operating as an astrocytic sensor of glutamate transmission in the nervous system, astrocytic mGluR5 is also subjected to regulation. Neuroinflammation, often observed in the context of nervous system injuries or neurological diseases, was shown to influence mGluR5 expression in astrocytes. The downregulation of mGluR5 was indeed replicated in vitro by exposing cultured astrocytes to selected inflammatory mediators [51,52].

The modulation of astrocyte responses to glutamate does not only operate through the regulation of mGluR5 expression, but also involves changes in the associated intracellular signaling. In astrocytes, the activation of mGluR5 is known to induce intracellular calcium oscillations, a characteristic feature of this receptor subtype [53]. Consistent with previous investigations, we have herein demonstrated that this specific signaling pattern is recapitulated in astrocyte cultures matured in the presence of a defined cocktail of growth factors. We previously demonstrated that changes in the expression of PKCε influenced the dynamics of mGluR5-induced intracellular calcium mobilization in astrocytes [54]. Additionally, our recent research on the regulation of mGluR5 signaling partners showed that neuroinflammation led to a significant alteration in RGS expression within astrocytes [39]. Specifically, RGS4, which is commonly seen as an actor in diverse nervous pathologies, was found to be upregulated after exposure to tumor necrosis factor α [26,32,55]. While the consequences of RGS4 modulation remain largely unexplored, the concurrent regulation of RGS4 and mGluR5 in astrocytic pathological conditions has prompted us to investigate whether mGluR5 signaling might be influenced by RGS4. This hypothesis is supported by experimental findings indicating a significant role of these two proteins in the mechanisms underlying neurological diseases. The functional interplay between RGS4 and mGluR5 has previously been demonstrated in neurons and specific brain structures [35,36]. However, this interaction has yet to be explored in glial cells, and therefore this question was herein addressed by focusing on the mGluR5 agonist-induced mobilization of intracellular calcium. Our findings reveal that the genetic or pharmacological manipulation of RGS4 influences the calcium signal associated with mGluR5 activation in astrocytes. In culture, individual cells exhibit distinct calcium signaling profiles following mGluR5 activation, characterized by either a sustained peak–plateau response or an oscillatory pattern [56].

On the one hand, the pharmacological manipulation of RGS4 activity using the synthetic selective and reversible allosteric inhibitor CCG 63802 profoundly modified the signaling triggered upon mGluR5 activation. It significantly decreased the number of oscillating cells, an effect that was reproduced through the lentiviral-mediated genetic downregulation of RGS4. Using the pharmacological inhibitor, the loss of an oscillatory pattern correlated with a reduction in the quantity of cells exhibiting a response to DHPG. Conversely, using the genetic approach, we observed a shift from the oscillating patterns to peak–plateau responses. This observation indicates that alterations in protein expression or activity might distinctly impact GPCR signaling. However, one cannot exclude that the inhibition of RGS4 at the employed concentration might influence other signaling partners and potentially induce off-target effects, as previously reported for other RGS4 inhibitors belonging to the same chemical series [33]. On the other hand, overexpressing RGS4 resulted in an increased percentage of cells showing an oscillatory response profile. The conversion from peak–plateau responses to an oscillatory pattern while upregulating RGS4 and vice versa is consistent with the concept that RGSs are negative modulators of GPCR signaling. Thus, an increase in RGS4 should promote the uncoupling of mGluR5 with the G_q_-type G protein, enabling a return to baseline cytosolic calcium levels, which leads to calcium transients during continuous stimulation. On the other hand, downregulating RGS4 would prevent the termination of the signal, thus leading to a sustained calcium response. The appearance of a more pronounced descending slope in the peak–plateau-responding cells upon RGS4 overexpression also corroborates this interpretation.

The dynamic control of agonist-evoked calcium responses at GPCRs by RGS4 was previously reported as this protein was proposed to be required for the generation of an oscillatory pattern [57]. Both the scaffolding and catalytic roles of RGS4 in cell microdomains appear to support the cell sublocalization and dynamic pattern of calcium signaling. With respect to the response to muscarinic receptor agonists, the knockdown of RGS4 disrupted calcium oscillations, converting the signal into a sustained plateau [58]. While the physiological implications of a calcium oscillation pattern remain under investigation, it is expected that the precise regulation of calcium signaling plays a pivotal role in encoding diverse biological events [15,59,60]. These include cell proliferation, transmitter release and the activation of transcription factors. Intracellular calcium elevations in astrocytes participates in the control of gliotransmitter release, which is essential for the glial-supported modulation of synaptic activity and plasticity [10,61]. While many studies often propose that the control of oscillations is driven by the concentration of ligands or directly by the density of the receptor, studies have highlighted the importance of receptor signaling partners such as RGSs or PKCε [13,54,60]. In this context, our previous studies on astrocytes derived from a rodent model of ALS demonstrated that calcium signaling patterns associated with mGluR5 activation could influence the regulation of glutamate uptake in astrocytes [62].

## 5. Conclusions

GPCRs are the targets of about one-third of the currently commercialized drugs worldwide [63,64,65]. In diverse pathological conditions, including in diseases of the nervous system, the dysregulation of these receptors or their associated signaling pathways is observed. However, most of the agonists and antagonists acting on GPCRs show limitations that include a loss of efficacy when used for a prolonged period of time, caused by desensitization, and unavoidable side effects, due to their widespread distribution in the body and the lack of selectivity of those drugs [17]. For this reason, the identification of alternative molecular targets is presently privileged and, among these, GPCR signaling partners receive considerable attention. Indirect signaling modulators such as RGS4 inhibitors, used in combination with receptor ligands, could pave the way for new therapeutic approaches with improved efficacy and selectivity. The present demonstration of the influence of RGS4 in intracellular signaling triggered by mGluR5 activation in cultured astrocytes warrants further exploration within an in vivo context. It is, however, noteworthy that the herein-used RGS4 pharmacological inhibitor has shown antiallodynic-like and antihyperalgesic-like properties when administered into the spinal cord of rodents undergoing neuropathic nerve lesions [32,66]. The antinociceptive responses to both opioid and cannabinoid receptor ligands appeared to be modulated when manipulating RGS4 [67]. Similarly, considering the established role of mGluR5 in modulating nociception [68,69,70], this particular subtype of glutamate receptor should also be envisaged as a potential candidate for functional modulation by RGS4, not only in neurons but also in glial cells [29].

## Figures and Tables

**Figure 1 cells-13-00291-f001:**
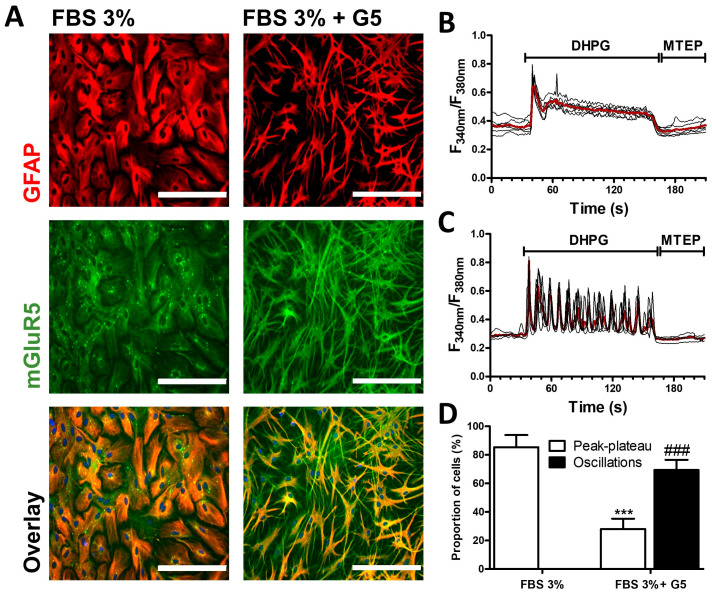
mGluR5 expression and activity in rat cortical astrocyte cultures. After 7 days of maturation in FBS 3% alone or in combination with the growth factor supplement G5, astrocytes were immunostained for GFAP (red) and mGluR5 (green) (bar 200 µm) (**A**). The bottom panels represent the overlay of both stains. Astrocytes were exposed to DHPG (50 µM) for 120 s, then to MTEP (10 µM) for 60 s, and changes in the cytosolic calcium concentration were monitored using Fura2. Representative traces indicating a Fura2 fluorescence ratio from 4 to 8 different cells are shown for astrocyte cultures matured in FBS 3% alone (**B**) or supplemented with G5 (**C**). Red traces represent the average of the signal measured for these cells. Panel (**D**) illustrates, for each condition, the occurrence of different types of calcium signaling profiles. Descriptive statistics (mean, SEM) were obtained from at least three different experiments. Statistical tests were performed through a hierarchical mixed logistic regression, fitted by maximum likelihood, with culture condition as fixed and experiment repetition as the random effect. Comparisons between fixed effects were obtained using Wald *z*-test statistics (**D**) (*** *p* < 0.001, ### *p* < 0.001).

**Figure 2 cells-13-00291-f002:**
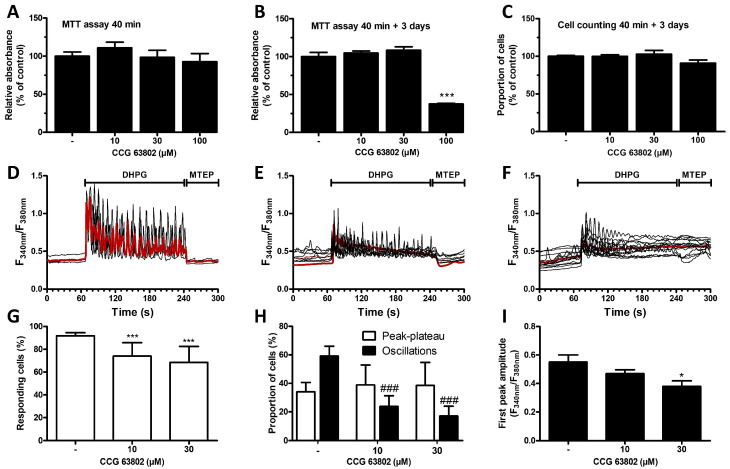
Influence of the RGS4 inhibitor CCG 63802 on cell viability and mGluR5-associated calcium signaling in astrocytes. Astrocytes were incubated for 40 min with the specific RGS4 inhibitor (CCG 63802, 10, 30 and 100 µM) and an MTT assay was performed either directly (**A**) or 3 days later (**B**). Cell viability was also assessed by cell counting 3 days after a 40 min RGS4 inhibition (**C**). Astrocytes were exposed to DHPG (50 µM) for 120 s, then to MTEP (10 µM) for 60 s, and changes in cytosolic Ca^2+^ concentration were monitored using Fura2. Representative traces indicating a Fura2 fluorescence ratio from 4 to 8 different cells are shown for naive astrocyte cultures or for cells preincubated for 40 min with CCG 63802 (control—(**D**), 10 µM—(**E**), 30 µM—(**F**)), and the red traces represent the average of the signal measured for these cells. Panels (**G**,**H**) illustrate, for each condition, the proportion of responding cells and the occurrence of different types of calcium signaling patterns, respectively. Panel (**I**) represents the first peak amplitude for each condition. Descriptive statistics (mean, SEM) were obtained from at least three different experiments. Statistical tests were performed through a one-way ANOVA followed by Student pairwise comparisons (**A**–**C**,**I**) or through a hierarchical mixed logistic regression, fitted by maximum likelihood, with treatment as fixed and experiment repetition as the random effect. Comparisons between fixed effects were obtained using Wald *z*-test statistics (**G**,**H**) (* *p* < 0.05, *** *p* < 0.001, ### *p* < 0.001).

**Figure 3 cells-13-00291-f003:**
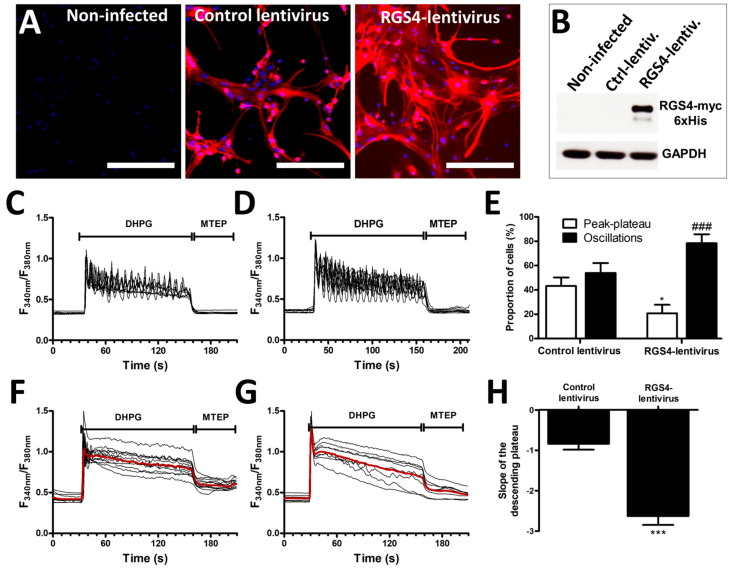
Modulation of mGluR5-associated calcium signaling by RGS4 overexpression in astrocytes. Astrocyte cultures were infected with a control lentivirus or as RGS4-lentivirus, both carrying an mCherry expression cassette, allowing for the detection of infected cells (bar 400 µm) (**A**). RGS4 expression was confirmed by immunoblotting (**B**). Astrocytes were exposed to 50 µM DHPG for 120 s, then to 10 µM of MTEP for 60 s, and changes in the cytosolic calcium concentration were monitored using Fura2. Representative traces indicating a Fura2 fluorescence ratio from 4 to 8 different cells are shown for astrocyte cultures infected with control lentiviruses (**C**) or with lentiviruses encoding RGS4 (**D**). Panel (**E**) illustrates the proportion of cells showing calcium oscillations and peak–plateau responses for each condition. Panels (**F**,**G**) are representative of the signaling recorded in cells responding with a peak–plateau profile after infection with the control and RGS4 lentiviruses, respectively, and the red traces represent the average of the signal measured for these cells. The histogram in (**H**) shows the slope depicting the decay of the plateau from these cells. Descriptive statistics (mean, SEM) were obtained from at least three different experiments. Statistical tests were performed through a hierarchical mixed logistic regression fitted by maximum likelihood, with lentiviral infection as fixed and experiment repetition as the random effect, along with Wald *z*-test statistics for comparisons between fixed effects (**E**) or a one-way ANOVA followed by Student pairwise comparisons (**H**) (* *p* < 0.05, *** *p* < 0.001, ### *p* < 0.001).

**Figure 4 cells-13-00291-f004:**
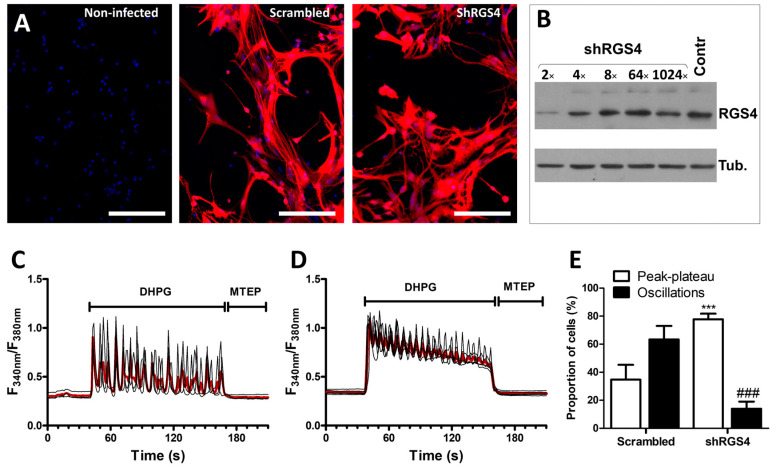
Modulation of mGluR5-associated calcium signaling by RGS4 downregulation in astrocytes. Astrocytes were infected with a scrambled shRNA or an shRNA directed against RGS4 (shRGS4), both carrying an E2-Crimson expression cassette, allowing for the detection of infected cells (bar 400 µm) (**A**). RGS4 expression was evaluated by immunoblotting using serial dilutions of the lentivirus suspension (**B**). Astrocytes were exposed to DHPG (50 µM) for 120 s, then to MTEP (10 µM) for 60 s, and changes in the cytosolic calcium concentration were monitored using Fura2. Representative traces indicating a Fura2 fluorescence ratio from 4 to 8 different cells are shown for astrocyte cultures infected with scrambled lentiviruses (**C**) or with shRGS4 lentiviruses (**D**). Red traces represent the average of the signal measured for these cells. Panel (**E**) illustrates the proportion of cells showing calcium oscillations and peak–plateau responses. Descriptive statistics (mean, SEM) were obtained from at least three different experiments. Statistical tests were performed through a hierarchical mixed logistic regression, fitted by maximum likelihood, with lentiviral infection as fixed and experiment repetition as the random effect. Comparisons between fixed effects were obtained using Wald *z*-test statistic (**E**) (*** *p* < 0.001, ### *p* < 0.001).

## Data Availability

The data presented in this study are available from the corresponding author on reasonable request.

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
