# Peer review of "Modulation of Type 5 Metabotropic Glutamate Receptor-Mediated Intracellular Calcium Mobilization by Regulator of G Protein Signaling 4 (RGS4) in Cultured Astrocytes"

_cells, 2024, doi:10.3390/cells13040291_

Round 1
Reviewer 1 Report
Comments and Suggestions for Authors
Review Report for "Modulation of mGluR5-mediated intracellular calcium mobilization by regulator of G protein signaling 4 (RGS4) in cultured astrocytes"
Summary: This paper investigates the role of RGS4 in regulating intracellular calcium signaling mediated by type 5 metabotropic glutamate receptor (mGluR5) in cortical astrocytes. The authors used a combination of pharmacological and genetic approaches to manipulate RGS4 activity in cultured astrocytes and examined the effects on mGluR5-mediated calcium signaling. In the first part of the study, the authors examined the influence of astrocyte maturation conditions on the expression and function of mGluR5. They found that astrocytes exposed to a standardized combination of growth factors (supplement G5) adopted a star-shaped morphology of stellate astrocytes and exhibited increased expression and functional properties of mGluR5 compared to astrocytes maintained in the FBS 3% condition. Next, the authors investigated the effect of RGS4 inhibition on mGluR5-mediated calcium signaling in G5-matured astrocytes. They found that exposure to the RGS4 specific inhibitor CCG 63802 increased the amplitude and duration of calcium transients in response to mGluR5 activation. In the third part of the study, the authors examined the effect of RGS4 overexpression on mGluR5-mediated calcium signaling in G5-matured astrocytes. They used a lentiviral-based approach to increase RGS4 expression and found that RGS4 overexpression reduced the amplitude and duration of calcium transients in response to mGluR5 activation. Overall, the study suggests that RGS4 plays an important role in regulating mGluR5-mediated calcium signaling in astrocytes. The authors discuss the potential implications of altered RGS4 expression in diverse pathologies, including neurological disorders where glutamate levels are dysregulated.
General Comments: The paper presents an interesting study on the role of RGS4 in regulating intracellular calcium signaling mediated by mGluR5 in cortical astrocytes. The authors provide a detailed description of the experimental methods and results, and the paper is well-written and organized. However, there are a few areas where the authors could improve the clarity and rigor of their study.
Specific Comments: The authors should provide more information on the rationale for using the specific shRNA sequence targeting rat RGS4 transcript. What evidence supports the specificity and efficacy of this shRNA sequence in knocking down RGS4 expression in astrocytes?
The authors should provide more details on the statistical analysis of their data. For example, what specific statistical tests were used to compare the different treatment groups, and what were the p-values for each comparison? The authors should also provide more information on the sample size and variability of their experiments.
The authors should discuss the potential limitations and caveats of their study. For example, what are the potential off-target effects of lentiviral-mediated RGS4 overexpression or knockdown? How do the in vitro findings of this study relate to the in vivo function of RGS4 in astrocytes and other cell types?
The authors should provide more context and discussion on the potential implications of their findings for diverse pathologies. For example, how might altered RGS4 expression or activity contribute to neurological disorders such as epilepsy, schizophrenia, or Alzheimer's disease? What are the potential therapeutic implications of targeting RGS4 in these disorders?
L36 and L36: Citation needed on works highlighting the role(s) of “system Xc”
-------------------------------------------------------------------------------------
L47-48
good elaboration on the coorperation between mGluRs and Gq/11, however a little more reason ruling out PKC (Calcium-dependent protein kinase) which can also affect gliotransmitter release and focusing only on mGluR5 should be highlighted?
-----------------------------------------------------------------------------------
L84 – 85
Although you have pointed out that there are currently no published research articles directly investigating the specific functional consequences of an interaction between RGS4 and mGluR5 in astrocytes, few published papers such as:
· Link: https://www.sciencedirect.com/science/article/pii/S1054358920300077
utilizes computational models to suggest a potential physical interaction between RGS4 and mGluR5 and hypothesizes its impact on astrocytic Ca2+ signaling.
· Link: https://www.ncbi.nlm.nih.gov/pmc/articles/PMC6588349/
and this also, provides evidence for a direct physical interaction between RGS4 and the G protein-coupled domain of mGluR5 through biophysical techniques. Might be interesting to look at and included in your reference pool.
-------------------------------------------------------------------------------------
L122: Did you try with different dye like Fluo-4 and obtain similar calcium signals? or whats your reason for using Fura-2? I know the choice could be experimental design requirements but the Fura-2 reported to be slower responding than Fluo-4, has lower signal-to-noise ratio and photobleaches easily. ? I am just curious to know your motivation for using the Fura-2?
L228 -229: Will the cytosolic calcium concentration signals or peak values change significantly if different dye was used?
-------------------------------------------------------------------------------------
L171: How similar or different will your results be if the samples are from humans instead of rats if yes, how different are the human model compared with the animal model? You investigated both down-expression and over-expression cases. Could there be any possibility of a switch because of human data – do we have any existing mGluR5 work on human astrocytes?
-------------------------------------------------------------------------------------
L289: ...changes in cytosolic calcium concentration were measured or recorded using Fura-2. The word “measured or recorded” is missing. Typo!
-------------------------------------------------------------------------------------
L317: … resulted in a substantial…. And not resulted in as substantial… ? Typo “as” instead of “a”
-------------------------------------------------------------------------------------
L333 – 334: what happens to the cytosolic calcium signals if astrocytes were to be exposed for longer than 120s or 60 s periods? I know you can expose astrocytess to DHPG (50μM) for longer than 120 seconds and then to MTEP (10μM) for longer than 60 seconds while observing changes in cytosolic calcium concentration using Fura-2. What are some of the limitations from experimental design point of view or respect to the Fura-2 consideration?
By carefully considering these aspects and optimizing your experimental design, you can successfully expose astrocytes to DHPG and MTEP for extended periods and gain valuable insights into their effects on cytosolic calcium dynamics using Fura-2.
-------------------------------------------------------------------------------------
L426: It would be interesting to see the extension of your work when RGS4 inhibitors, are used
in combination with receptor ligands or ion channels?
-------------------------------------------------------------------------------------
Overall Assessment: This is a well-executed and informative study on the role of RGS4 in regulating mGluR5-mediated calcium signaling in astrocytes. However, the authors could improve the clarity and rigor of their study by addressing the specific comments outlined above.
Comments on the Quality of English LanguageMinor editing of English language required.
Reviewer 2 Report
Comments and Suggestions for Authors
The paper by dr. Beckers and colleagues reports interesting findings about the role of RGS4 in modulating mGluR5-mediated responses in cerebral astrocytes. This investigation provides solid data contributing to the knowledge of signaling regulation at the tripartite synapse and indicates possible targets for pharmacological treatments of nervous diseases. I have only a few observations that the Authors may take into considerations.
Lines 31-35: A reference summarizing the general properties of astrocytes should be cited.
Lines 211-213: Immunofluorescence is not a quantitative technique, therefore, in principle, it cannot be used to compare the amount of a protein in different samples unless some kind of densitometric analysis is performed. Even less affordable is to assess gene expression. The differences in immunostaining can be easily appreciated by eye, but I don’t think you can say that they demonstrate increased mGluR5 expression.
- After the assessment of the differences between different culture conditions (with or without G5 supplement), the experiments were performed on G5-matured cultures. Since this analysis occupies a whole paragraph (3.1) in the results, the observed differences should be discussed (although briefly) and the reasons for choosing the G5 cultures should be explained.
Line 258: Fig. 2 does not show only calcium signaling, but also MTT data.
Figs 3 and 4: The comparisons are made between control lentivirus and RGS4-lentivirus (fig. 3) or between scrambled and ShRGS4. In addition, on line 301, it is said that “Infecting astrocyte cultures modestly influenced the calcium signalling induced by DHPG”. I think it is necessary to also show data from non-infected cultures (perhaps as supplementary material), just to show that control or scrambled cultures are not altered.
Comments on the Quality of English LanguageQuality of English is fine with me (only minor editing needed)
Reviewer 3 Report
Comments and Suggestions for Authors
The manuscript by P. Beckers et al. describes the functional effect of RGS4 on the mGluR5-mediated calcium signaling in cultured astrocytes. This manuscript is well-written, clearly discussed and the results are presented clearly.
I only have minor comments:
Introduction:
- line 38: (mGluR), a G protein-coupled receptor (remove s family)
- line 47: preferentially couples...instead of preferential couples
Materials and Methods:
- There seems to be no reference for the CCG 63802 compound. Is it commercially available or not? Please add to the M&M section.
- line 108: 24-well plates
- line 114: explain GFAP
Results:
- line 283: refer to Figure 3B
- line 289: calcium concentration were....using Fura2. Complete the sentence.
- line 319: add a space between immunoblotting and using.
- Figure 4, panel B: explain 2x, 4x...Also, what is the shRGS4 dilution actually used for the Ca2+ signaling experiments?
Discussion
- line 378: a astrocytes. Remove a.
